# PeerJ

# An interactive three dimensional approach to anatomical description— the jaw musculature of the Australian laughing kookaburra (*Dacelo novaeguineae*)

Michelle R. Quayle[1], David G. Barnes[2,3,4], Owen L. Kaluza[5] and Colin R. McHenry[1]

[1] Department of Anatomy and Developmental Biology, School of Biomedical Sciences, Monash University, Clayton, VIC, Australia
[2] Monash Biomedical Imaging, Monash University, Clayton, VIC, Australia
[3] VLSCI Life Sciences Computation Centre, Carlton, VIC, Australia
[4] Clayton School of Information Technology, Monash University, Clayton, VIC, Australia
[5] Monash e-Research Centre, Monash University, Clayton, VIC, Australia

## ABSTRACT

The investigation of form-function relationships requires a detailed understanding of anatomical systems. Here we document the 3-dimensional morphology of the cranial musculoskeletal anatomy in the Australian Laughing Kookaburra *Dacelo novaeguineae*, with a focus upon the geometry and attachments of the jaw muscles in this species. The head of a deceased specimen was CT scanned, and an accurate 3D representation of the skull and jaw muscles was generated through manual segmentation of the CT scan images, and augmented by dissection of the specimen. We identified 14 major jaw muscles: 6 in the temporal group (*M. adductor mandibulae* and *M. pseudotemporalis*), 7 in the pterygoid group (*M. pterygoideus dorsalis* and *M. pterygoideus ventralis*), and the single jaw abductor *M. depressor mandibulae*. Previous descriptions of avian jaw musculature are hindered by limited visual representation and inconsistency in the nomenclature. To address these issues, we: (1) present the 3D model produced from the segmentation process as a digital, fully interactive model in the form of an embedded 3D image, which can be viewed from any angle, and within which major components can be set as opaque, transparent, or hidden, allowing the anatomy to be visualised as required to provide a detailed understanding of the jaw anatomy; (2) provide a summary of the nomenclature used throughout the avian jaw muscle literature. The approach presented here provides considerable advantages for the documentation and communication of detailed anatomical structures in a wide range of taxa.

Corresponding author
Michelle R. Quayle,
quayle.michelle@gmail.com

**Peer**J ______________________________________________

## INTRODUCTION

The relationship between form and function is a central concept in biology (*Lauder et al., 1995*; *Russell, 1916*; *Thompson, 1917*); it determines the extent to which the anatomy of an organism permits specific behaviours and ecology. Investigating form-function relationships requires a detailed understanding of anatomical systems, i.e., the discipline of anatomy and morphology. For functional questions concerning feeding ecology, the relevant anatomical system is the feeding apparatus; for the majority of vertebrates, this consists of the jaws. The jaw is a functional unit, consisting of a rigid framework—the skull bones of the cranium and mandible, operated by a system of jaw muscles. The skull bones must be able to withstand the forces of catching a struggling prey item, whilst transferring the bite force generated from the jaw muscles to hold, kill and process the prey without sustaining fracture. Fundamentally, the mechanical limits of the jaw's ability to bite the prey item determines the type and size of prey that can be taken (*Ferrara et al., 2011*; *Herrel, O'Reilly & Richmond, 2002*; *Wroe, McHenry & Thomason, 2005*).

Form-function relationships can be investigated using mechanistic techniques, such as computational biomechanics, and/or quantitative approaches, such as statistics and morphometrics. For biomechanical modelling, the jaw musculature is an important component of the loads that act upon the skull during feeding and other behaviours. During biting, **isotonic** contraction of the adductor musculature imposes reaction forces on the prey (or other item), whilst shaking or twisting of the food requires **isometric** contraction of the jaw muscles to firmly hold the prey. During each of these activities, the activation of the jaw musculature imposes tensile forces to attachment areas which can result in loads at least as high as the loads being applied to the prey (*Herring et al., 2001*). Therefore, to simulate the jaw as closely to the actual system as possible it is important to include an accurate representation of the jaw musculature in the model.

The following description of the jaw musculature of the Australian laughing kookaburra (*Dacelo novaeguineae*) was stimulated by the need to incorporate detailed anatomical data into high resolution 3D finite element models as part of an analysis of skull biomechanics in 8 species of kingfisher (Alcedines) (*Quayle, 2011*; MR Quayle, M Mahony, P Clausen, MR McCurry, CW Walmsley, CR McHenry, unpublished data). *Dacelo novaeguineae* was chosen as a representative species of kingfisher for anatomical description as it is a common species in our location meaning specimens are readily available and it has a large body size compared to other species of kingfishers; allowing for the small jaw muscles to be more easily detectable through imaging and dissection techniques. This is the first time that the jaw musculature of this species has been documented.

### Avian and kingfisher cranial anatomy

A review of the literature indicates that descriptions of avian jaw anatomy are limited, with studies of only a few species, use of two-dimensional drawings, and complicated written descriptions. A major obstacle is the use of differing terminology (*Holliday & Witmer, 2007*). Papers that incorporate higher levels of detail include Lautenschlager, Bright and Rayfield's description of the common buzzard jaw musculature (*2013*), Holliday and

Witmer's description and illustration of the albatross jaw musculature (*2007*), Elzanowski's description of the tinamous jaw (*1987*), Zusi and Storer's pied-billed grebe jaw study (*1969*), Richards and Bock's comparison of Hawaiian honeycreeper feeding apparatus (*1973*), and Zweers' description of mallard myography (*1974*). From these descriptions it is evident that avian jaw muscle anatomy differs in relative and absolute size of the various muscles, and in the position and extent of attachments. Consequently, we determined that the jaw musculature in kingfishers should not be reconstructed on the basis of anatomical descriptions of other families.

The jaw muscles of several species of kingfisher (though not *Dacelo novaeguineae*) have been described by *Burton (1984)*; however his illustrations are difficult to interpret, with little information identifying which part of the skull he is referring to in the figures. Burton's description did, however, reveal several variations in muscle origin and insertion areas between the kingfisher species he studied. This observation reinforced the need to document the kookaburra anatomy independently.

The lack of detailed published data on the kookaburra jaw musculature meant that firsthand anatomical information was required to accurately model the jaw system. This information was obtained through the dissection of the head of an adult laughing kookaburra (*Dacelo novaeguineae*). The kookaburra was CT scanned prior to dissection, and this scan data was used in conjunction with the dissection information to create a three-dimensional computer model of the skull and jaw muscles of *D. novaeguineae*.

## A 3D approach to anatomical description

Anatomy is inherently three-dimensional, but traditional descriptions of anatomy use a combination of text and 2D picture modes. New technologies in data collection, assembly, visualisation and communication, are offering significant new opportunities for improving the study, documentation and publication of anatomy, particularly given the proliferation of 3D imaging capabilities.

The following description adopts and exemplifies a three-dimensional approach to displaying anatomical systems through the use of (1) three-dimensional Adobe Portable Document Format (3D PDF) figures, and (2) 3D-enabled web pages. The 3D PDF medium allows an interactive 3D model to be inserted into an electronic document in PDF format, meaning that specialised software is not required to view the model other than the freely available Adobe Reader (desktop version), and permanently embedding the figure directly in the document file. This method of interactive visualisation is becoming increasingly popular, with publications in biomedical science (*Kumar et al., 2009*; *Selvam, Vasilyev & Wang, 2009*), astrophysics (*Barnes & Fluke, 2008*), invertebrate (*Baeumler, Haszprunar & Ruthensteiner, 2008*; *Neusser, Heß & Schrödl, 2009*; *Ziegler et al., 2008*) and vertebrate anatomy (*de Bakker et al., 2012*; *Holliday et al., 2013*; *Ruthensteiner & Heß, 2008*; *Lautenschlager, Bright & Rayfield, 2013*) and paleontology (*Knoll et al., 2012*; *Lautenschlager, 2013*). In addition, free, open source software that provides a pathway for the generation of 3D PDF figures, especially for use in science, is now available (*Barnes et al., 2013*).

The 3D meshes produced for the 3D PDF were also used to generate a model available for viewing in web pages. Web-based 3D visualisation is becoming more common with the growing number of browsers supporting accelerated 2D and 3D graphics, especially via the WebGL standard (http://www.khronos.org/webgl); recent examples include applications in bioinformatics (*Pettit & Marioni, 2013*), neuroinformatics (*Ginsburg et al., 2011*), and surgical teaching (*Birr et al., 2013*). Like 3D PDF, web-based 3D visualisation enables the interactive manipulation of a 3D model without requiring specialised software other than a modern web browser.

Here we present a detailed description of the 3D osteology and jaw musculature of *D. novaeguineae*, and provide a summary of the avian jaw muscle synonymies. We aimed to demonstrate and exploit the potential of 3D viewing technology in the anatomical sciences by producing a 3D model that shows the position of the kookaburra jaw muscles on the skull relative to one another; the model can be rotated and superficial muscles hidden or made transparent to allow better visualisation of the deeper anatomy. While the 3D model allows for a detailed descriptive model in a single figure-frame, it is only available for viewing in digital format. For this reason, still images suitable for 2D print media are also included.

## MATERIALS & METHODS

### Specimen

A fresh, deceased adult laughing kookaburra (*Dacelo novaeguineae*), obtained from Australia Museum's Ornithological collection (specimen no. AM BF2162) was used in the following description. This specimen had died in the care of wildlife rescuers as a result of an infection on the right elbow, although the head and skull did not show any obvious pathologies and thus were deemed to illustrate normative anatomy.

### Data acquisition and processing

The kookaburra was scanned in a clinical computed tomography (CT) scanner (Toshiba Aquilion 64) at the Newcastle Calvary Mater Hospital, at 120 kV, 75 mA and using a slice thickness of 0.5 mm. The resulting DICOM images were imported into MIMICS v11 (MATERIALISE, Belgium) segmentation software, wherein the skull of the kookaburra was digitally separated from the soft tissue by combining a scalar threshold with manual selection tools; the high density of the bone compared to the surrounding soft tissue allowed for quick and simple segmentation of the skull by applying a threshold based on the grey scale pixel values representative of the bone. A 3D model of the cranium and mandible was then created from this segmentation. This digital skull model provided a scaffold to illustrate the muscle attachment points and muscle geometry determined in the dissection (Fig. 1).

Segmentation of the jaw muscles proved more difficult due to the small size of the muscles, the low resolution of the scanner and the use of CT (without staining for contrast) rather than MRI. While the soft tissue making up the kookaburra jaw muscles was visible in the CT scan data, and the bellies of the larger muscles could be differentiated and separated

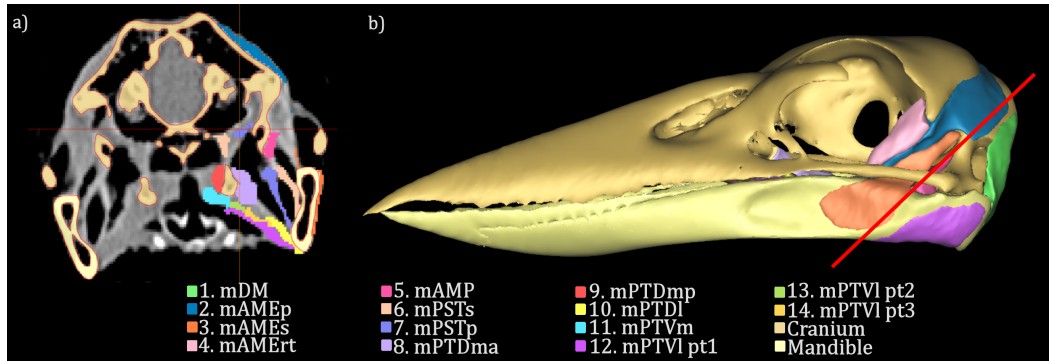

**Figure 1 Segmentation of the CT data.** The CT scan data for the *Dacelo novaeguineae* head was imported into the segmentation software MIMICS (MATERIALISE, Belgium) (A). The skull was segmented out using thresholding tools to select the regions of bone from the scan images, while the muscles were manually segmented. Segmentation is indicated by the coloured regions on the CT image. The segmented regions on each CT image slice were then joined using Mimics modelling tools to create a 3D model of the *D. novaeguineae* skull and jaw muscles (B). The red line through the skull in (B) indicates the location of the CT slice in (A). Jaw muscles shown in this model are 1. *M. depressor mandibulae*, 2. *M. adductor mandibulae externus profundus*, 3. *M. adductor mandibulae externus superficialis*, 4. *M. adductor mandibulae externus rostralis temporalis post-orbital lobe*, 5. *M. adductor mandibulae posterior*, 6. *M. pseudotemporalis superficialis*, 7. *M. pseudotemporalis profundus*, 8. *M. pterygoideus dorsalis medialis anterior*, 9. *M. pterygoideus dorsalis medialis posterior*, 10. *M. pterygoideus dorsalis lateralis*, 11. *M. pterygoideus ventralis medialis*, 12. *M. pterygoideus ventralis lateralis Part 1*, 13. *M. pterygoideus ventralis lateralis Part 2* and 14. *M. pterygoideus ventralis lateralis Part 3*.

using MIMICS segmentation tools, the resolution of the scan was too low to accurately identify the smaller jaw muscles and the muscle attachment sites. For this reason, the soft tissue CT scan data was used alongside the dissection information to identify placement of the jaw muscles (Figs. 1 and 2).

The jaw muscles were dissected out beginning with the superficial layer. Each individual muscle was described using drawings, photographs and notes to illustrate its position, direction and attachment sites. Muscles of the hyoid and eye were not required for the biomechanical analysis and were not included in this study. The completed notes on kookaburra muscle anatomy were compared with previously published data on avian jaw musculature (*Baumel et al., 1993*; *Burton, 1984*; *Elzanowski, 1987*; *Holliday & Witmer, 2007*; *Zusi & Storer, 1969*) to determine the individual names of the jaw muscles observed. Muscle nomenclature used was based on *Holliday & Witmer (2007)* and *Burton (1984)*. Osteological nomenclature (Fig. 3) was based on *Baumel et al. (1993)*, *Zusi & Storer (1969)* and *Ghetie et al. (1976)*.

In conjunction with the CT scan data of the specimen, the illustrated dissection notes were used to digitally recreate a three dimensional representation of the kookaburra jaw musculature on the specimen's skull model (Fig. 2). This model was created using the CT editing software MIMICS to manually select areas of the CT image slices where the muscles were determined to occur. The CT slices were then joined in MIMICS to create a 3D model of each muscle which were individually exported as stereolithography (STL) format files.

**Peer**J ________________________

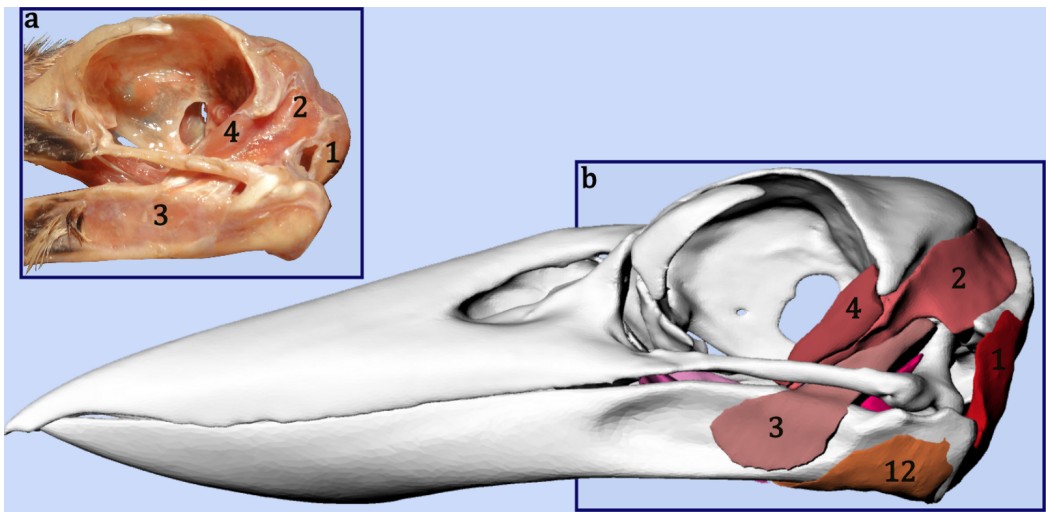

**Figure 2 Comparing the dissection to the digital model.** Each of the jaw muscles of the laughing kookaburra (*Dacelo novaeguineae*) were individually described and removed during the dissection (A). Notes, drawings and photographs from the dissection were used to identify the jaw muscles on the CT scan so that they could be digitally segmented and modelled (B). This diagram shows the placement of the muscles in the dissected specimen (A) and their digital representation (B). Muscle shown here are 1. *M. depressor mandibulae*, 2. *M. adductor mandibulae externus profundus*, 3. *M. adductor mandibulae externus superficialis*, 4. *M. adductor mandibulae externus rostralis temporalis post-orbital lobe* and 12. *M. pterygoideus ventralis lateralis part 1* (removed in dissection photo).

## 3D image production

### PDF

A total of 16 segmented surfaces (meshes)—the cranium, mandible, and 14 muscles—comprise the segmented jaw anatomy. The cranium was the largest and most detailed surface, and was remeshed and simplified in MeshLab version 1.3.2 (Visual Computing Lab–ISTI–CNR; http://meshlab.sourceforge.net) using a "Quadric Edge Collapse Decimation" to reduce its facet count from ∼635,000 to ∼100,000; this reduced the total model size without significant visible degradation. The meshes were converted (using MeshLab) to the widely-used Alias Wavefront ".OBJ" format and listed in a text file, one per line, with colon-separated fields for the part (anatomical) name, a group code, the filename, and the desired part colour expressed as an (r, g, b) triplet. The group code is used to assign parts to named components in a two-level hierarchy. The text file describing the model was read by a small, custom 3D visualisation program written using S2PLOT (*Barnes et al., 2006*), and the surfaces were loaded, coloured and displayed for inspection. Names were applied to the different meshes, and the export capability in the recent S2PLOT release version 3.2.1 (*Barnes et al., 2013*) was used to save the complete 3-D model to a Product Representation Compact (PRC) format file.

Additional lines in the model description file were used to: define standard views of the anatomy (left lateral, right lateral, anterior, posterior, dorsal and ventral); to select a basic lighting environment (a headlamp) and background colour; and to define the component group names. From the model description file, a custom views file suitable

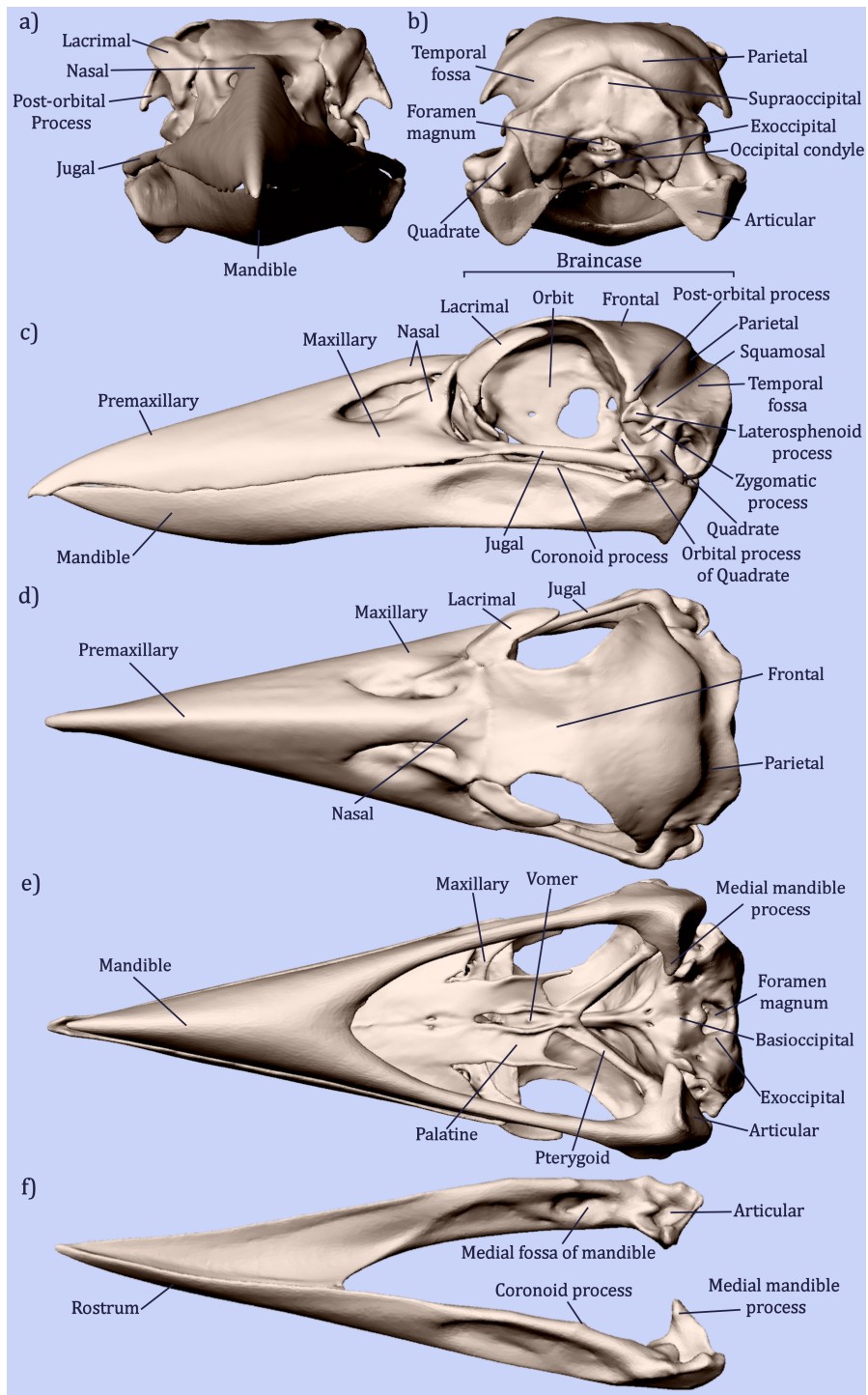

**Figure 3 Osteology of the avian skull.** Illustrating the bones and features of the avian skull on *Dacelo novaeguineae* in (A) anterior, (B) posterior, (C) lateral, (D) dorsal and (E) ventral views. F shows the features of the mandible in dorso-lateral view. Due to the fused nature of avian skull bones, the separation between the bones is not defined. Adapted from *Baumel et al. (1993)*, *Ghetie et al. (1976)* and *Zusi & Storer (1969)*.

for PDF embedding was generated, and JavaScript fragments to toggle the various component groups were generated. Following the workflow in *Barnes et al. (2013)*, the PRC model was embedded in a LaTeX-format document (Lamport, *LaTeX: A Document Preparation System (2nd Edition)*, 1994) and compiled into a PDF file with embedded 3D figure, incorporating the custom views file and JavaScript fragments and the standard 3D JavaScript code, included with S2PLOT version 3.2.1, which implements standard keystrokes such as cursor keys to rotate the model and the letter 'a' to toggle autospin mode.

## Web-based figure

Our web-based figure is based on the JavaScript *three.js* (http://threejs.org), public domain extensions to *three.js* (via http://learningthreejs.com), and E. Graether's *TrackballControls.js* (http://egraether.com). *Three.js* provides a simple entry point to visualisation of 3D scientific data on the Web by supporting the loading of several common 3D model data formats, and rendering the 3D model using WebGL (http://khronos.org/webgl) where available, and otherwise the 2D Javascript canvas API for devices with no WebGL support. The OBJLoader.js module provides a ready-made importer for the ubiquitous Alias Wavefront OBJ format. Using these components, we developed a custom JavaScript-based web page that reads the identical model description file used for assembling the 3D PDF figure, and constructs a web-based 3D visualisation, coloured as for the 3D PDF version, and offering the same capability to interactively rotate and zoom the model, and to toggle the model components.

## RESULTS

### Musculoskeletal anatomy

#### *Depressor muscles*

1. *M. depressor mandibulae* (mDM) has a fleshy origin on the squamosal region on the back of the braincase, just posterior to the opening of the ear. It becomes thicker as it extends around the jaw joint to insert over most of the posterior surface of the mandible articular bone (Figs. 4G and 4H).

#### *Temporal muscles*

The external adductor complex—*M. adductor mandibulae externus* (mAME)—has three parts:

2. M. *adductor mandibulae externus profundus* (mAMEp) originates along the length of the temporal fossa; the attachment extends from the sagittal midline of the cranium (Figs. 4G and 4H) to the region just dorsal to the zygomatic process and along the posterior edge of the post orbital process. *M. adductor mandibulae externus profundus* inserts onto the coronoid process of the mandible posterior to the insertion of *M. pseudotemporalis superficialis* (Figs. 4A, 4B and 4J).

3. *M. adductor mandibulae externus superficialis* (mAMEs) has a narrow origin located on the lateral side of the zygomatic process. It extends down around the medial side of the jugal where it fans out to insert widely and thinly over the dorso-lateral surface of the mandibular ramus (Figs. 4A, 4B and 4J).
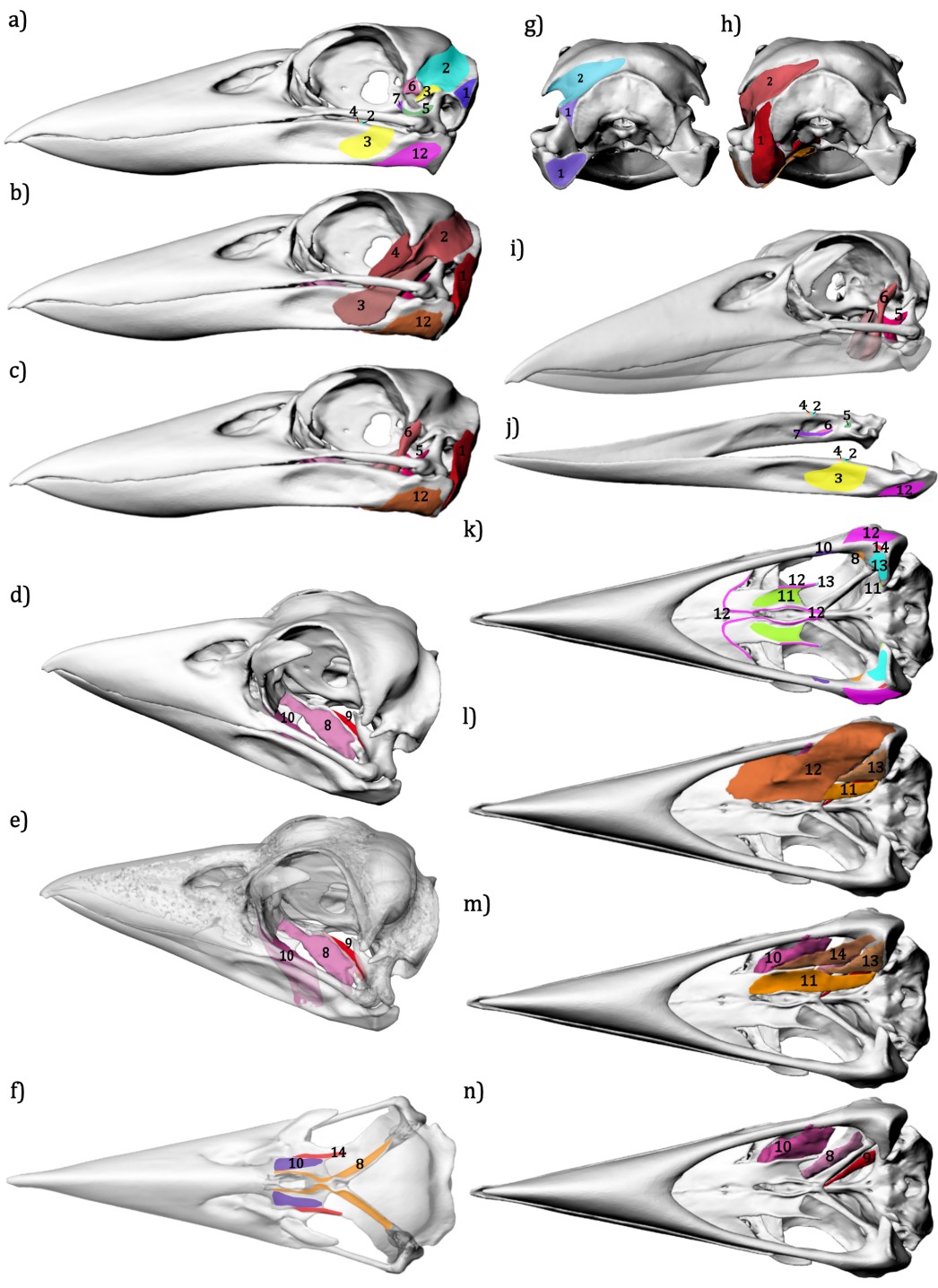

**Figure 4  Jaw musculature of the Australian Laughing Kookaburra *Dacelo novaeguineae.* (A) Lateral view showing the attachment areas of the jaw muscles, (B) the geometry of the jaw muscles and (C) deeper muscle geometry with muscles 2, 3 and 4 hidden. (D) Showing the geometry of the *M. pterygoideus dorsalis* muscles and (E) their position in a transparent skull. (F) Dorsal view of a transparent skull showing attachment sites of the *M. pterygoideus* muscles to the dorsal side of the palatine and pterygoid bones.  (continued on next page...)

**Figure 4 (...continued)**
(G) Attachment sites of muscles on the posterior side of the skull and (H) their geometry. (I) geometry of the *M. adductor mandibulae externus* and *M. pseudotemporalis* muscles, the skull is shown with a transparent mandible to display their medial attachment to the mandible. (J) Dorsolateral view of the muscle attachment sites on the mandible. (K) Muscle attachment sites on the ventral side of the skull, and the geometry of (L) superficial, (M) middle and (N) deep layers of ventral muscles. 1. *M. depressor mandibulae*, 2. *M. adductor mandibulae externus profundus*, 3. *M. adductor mandibulae externus superficialis*, 4. *M. adductor mandibulae externus rostralis temporalis post-orbital lobe*, 5. *M. adductor mandibulae posterior*, 6. *M. pseudotemporalis superficialis*, 7. *M. pseudotemporalis profundus*, 8. *M. pterygoideus dorsalis medialis anterior*, 9. *M. pterygoideus dorsalis medialis posterior*, 10. *M. pterygoideus dorsalis lateralis*, 11. *M. pterygoideus ventralis medialis*, 12. *M. pterygoideus ventralis lateralis Part 1*, 13. *M. pterygoideus ventralis lateralis Part 2* and 14. *M. pterygoideus ventralis lateralis Part 3*.

4. *M. adductor mandibulae externus rostralis temporalis post-orbital lobe* (mAMErt): This muscle is thought to be a member of the adductor mandibulae externus group. Burton (*1984*) labels it '*M. adductor mandibulae externus rostralis post-orbital slip*' or '*lobe*' in his illustrations but only briefly mentions it in the text. We can find no other literature that uses this term, but we retain its use here in lieu of further information on its possible homology. This muscle was found to have a fleshy origin on the anterior/medial side of the post-orbital process. It inserts onto the coronid process of the manidble, anterior to the insertion of *M. adductor mandibulae externus profundus* (Figs. 4A, 4B and 4J). It is clearly separate to the other parts of the external adductor complex (mAME).

5. *M. adductor mandibulae posterior* (mAMP) originates along the orbital process of the quadrate and the quadrate body (Figs. 4A, 4C, 4I and 4J). It inserts onto the dorsal side of the mandible, immediately antero-medial to the jaw joint.

### Pseudotemporalis (mPST) muscles

6. *M. pseudotemporalis superficialis* (mPSTs) has two sections (displayed here as one muscle). The medial part originates on the laterosphenoid process on the posterior wall of the orbit (Figs. 4A, 4C and 4I). The lateral part originates slightly higher in the depression lateral to the process. Both sections insert onto the medial side of the mandible where they fan out to cover most of medial fossa of the mandible, dorsal to the insertion of M. *pseudotemporalis profundus* (Figs. 4I and 4J).

7. *M. pseudotemporalis profundus* (mPSTp) originates from the medial point of the quadrate orbital process (Figs. 4A and 4I). The medial section of the muscle is thick and fleshy, while that lateral side consists of a thin sheet fanning out as it inserts over a large area on the ventral edge of the medial fossa of the mandible (Figs. 4I and 4J).

### Pterygoid (PT) muscles

8. *M. pterygoideus dorsalis medialis anterior* (mPTDma) originates along the entire length of the medial ridge of the palatine and onto the lateral edge of the pterygoid bone. It inserts onto the anterior end of the mandible's medial process (Figs. 4D, 4E, 4F and 4N).

9. *M. pterygoideus dorsalis medialis posterior* (mPTDmp) takes its origin along the medial side of the pterygoid bone, beginning from the joint of the palatine and pterygoid and inserting onto the anterior edge of the medial mandibular process (Figs. 4D, 4E and 4N).

10. *M. pterygoideus dorsalis lateralis* (mPTDl) has a fleshy origin on the anterior-most portion of the palatine and down onto the dorsal lateral surface of the palatine. This large muscle wraps down and around to insert onto the ventral/medial side of the mandible (Figs. 4D, 4E, 4F, 4K, 4M and 4N).

11. *M. pterygoideus ventralis medialis* (mPTVm) originates along the entire ventral surface of the palatine. It extends over the ventral side of the pterygoid bone to insert onto the anterior surface of the tip of the medial mandibular process (Figs. 4K, 4L and 4M).

The *M. pterygoidus ventralis lateralis* (mPTVl) muscle is formed from three parts:

12. *M. pterygoideus ventralis lateralis Part 1* (mPTVl pt1) is a large, flat, superficial muscle lying most ventral to all others. It originates along the lateral ventral ridge of the palatine and along the lateral side of the crest where the palatine meets the vomer by an aponeurotic sheet extending anteriorly until it merges with the skin of the palate. Posteriorly this muscle becomes thicker as it extends over the posterior end of the mandible to wrap around and insert onto the lateral side of the mandible, posterior to the attachment of *M. adductor mandibulae externus superficialis* (Figs. 4A, 4B, 4J, 4K and 4L).

13. *M. pterygoideus ventralis lateralis Part 2* (mPTVl pt2)—this muscle and *M. pterygoideus ventralis lateralis part 3* occur closely together and it is possible that they are different heads of the same muscle. Part 2 attaches to the lateral point of the palatine. It then extends back to fan out and attach over a reasonably large area on the ventral side of the entire medial mandibular process (Figs. 4K, 4L and 4M).

14. *M. pterygoideus ventralis lateralis Part 3* (mPTVl pt3) runs laterally adjacent to Part 2. It originates along the dorsal edge of the palatine, lateral to the palatine attachment of *M. pterygoideus dorsalis lateralis*. This muscle attaches to the ventral crest of the medial mandibular process (Figs. 4F, 4K and 4M).

### 3D visualisation

An interactive, 3D model of the jaw muscles described above is available in the Appendix and a web version of this model is available here http://cave2.github.io/websurfer/?model=examples/quayle_kookaburra.

## DISCUSSION

### Methodological issues

With the increasing use of CT as a primary means of collecting anatomical data for use in 3D modelling, it is important to note the limitations of this imaging modality. Whilst it is possible to collect the data required for the 3D modelling of the bones from CT alone, our experience with the specimen used in this study is that incorporating detailed muscle anatomy into a 3D model requires a combination of CT imaging and dissection. In CT, contrast between bone and soft tissue is high, which maximises the accuracy of segmentation protocols (both automatic algorithms and manual image processing) for segmenting bones. However, in the absence of fixing protocols (e.g., iodine potassium iodide—'Lugol's iodine'), contrast between muscles and other soft tissues is low and it is difficult to identify individual muscle bellies on the basis of CT data alone; both the perimysium and intermuscular septa are inconsistently visible. An additional limitation

is the comparatively low resolution of medical CT in imaging small animals; for the scans used in this study, maximum pixel resolution in each slice was 0.337 mm, which is a relatively large proportion of the maximum cranial diameter (44 mm), even for a large kingfisher species such as the kookaburra. Without the opportunity to use dissection to augment the 3D imaging of muscle anatomy we could not have produced the level of anatomical data presented here.

Whilst small-animal CT scanners (maximum field of view ~100 mm) or micro-CT (~20 mm) can deliver significantly higher resolution than clinical CT, these scanners are not as readily available as clinical CT scanners and we were not able to utilise one at the time of this study. These higher resolution CT scanners also cannot produce images that clearly differentiate muscle bellies without first preparing the specimen with an iodine stain, and require much longer scan times than a clinical CT.

It is possible that magnetic resonance imaging (MRI) can provide an alternative to dissection by producing high resolution images of the soft tissue, however MRI is not able to image bone well. A specimen can be both CT and MRI scanned and the resulting images co-registered to fit to the same 3D space co-ordinates, so that CT can be used to model the skeletal elements, while MRI can provide muscle detail. At the time of this study an MRI scanner and co-registration code was not available to us.

## Nomenclature

Determining the correct names of particular muscles identified in the kookaburra based on previous avian jaw anatomy literature was a complicated process. For many taxonomic groups, anatomical descriptions have ranged across decades, languages, and authors; the issues we discovered with respect to non-standard nomenclature are not restricted to kingfishers. Muscle groups, particularly the *M. adductor mandibulae externus* and *M. pterygoideus dorsalis* muscles were separated into several sections in some studies (*Burton, 1984*; *Richards & Bock, 1973*), or treated as a whole in others (*Holliday & Witmer, 2007*). Within this study, questions remain regarding the extent to which some muscle bellies should be regarded as individual muscles, or be considered as a part of a larger muscle. Specifically, is *M. adductor mandibulae externus rostralis temporalis post-orbital lobe* part of one of the other *M. adductor mandibulae externus* muscles (with the only reference found of a separate muscle in this area recorded by *Burton (1984)*; and should *M. pterygoideus ventralis lateralis Part 2* and *Part 3* be considered as separate muscles to *M. pterygoideus ventralis lateralis Part 1*?

Nomenclature used in this study was based mainly on *Burton*'s (*1984*) study as it includes kingfisher anatomy and uses similar terminology to Baumel et al.'s Nomina Anatomica Avium (*1993*). However, temporal muscle nomenclature was based upon *Holliday & Witmer (2007)*. *Baumel et al. (1993)* and *Burton (1984)* each separate *M. adductor mandibulae externus* into three parts of mAME rostralis, mAME ventralis and mAME caudalis, but we found no clear evidence of 5 separate heads for this muscle in *D. novaeguineae*. Holliday and Witmer's nomenclature for the external adductor complex was more similar to the anatomy visible in our specimen.

There is a lack of consistency in the way in which muscle names are written and abbreviated in text of previous literature. In an attempt to develop a standardised format for writing muscle names and abbreviations, we use the following format:

1. When writing the muscle name in text, we use an uppercase 'M.' (contraction for Musculus) at the start of the name, following Baumel et al.'s Nomina Anatomica Avium (*1993*).

2. When writing the muscle name in full within text, we italicise the name; hence, *M. pseudotemporalis superficialis*. The muscle name is more easily distinguished visually from the surrounding text, making the text easier to read.

3. When abbreviating the muscle (e.g., for figure labels), we use lower case for the 'm' and upper case for the abbreviation of the muscle name; hence, mPST. This format visually highlights the informative part of the label.

4. Parts of the muscle are written in lower case, hence mPSTs and mPSTp. This makes it easier to identify the muscle under discussion, avoiding the potential confusion that can arise between muscles in the external adductor complex if the abbreviation for the part is written in upper case—*Holliday & Witmer (2007)* use mAMEP and mAMP for the *M. adductor mandibulae externus profundus* and the *M. adductor externus posterior* respectively. We abbreviate these muscles as mAMEp and mAMP.

Table 1 lists the avian jaw muscle nomenclature that we have used in this paper, the respective abbreviations for use in labels, and the synonymies in selected literature. Note that some of these published nomenclatures were presented as anatomical illustrations rather than written descriptions, making it difficult to accurately determine exact muscle synonymy.

## The role of 3D images in communicating anatomical information

Descriptive anatomy is the foundation for understanding functional anatomy. The recent surge in biomechanical studies of form-function questions in a range of taxa means descriptive anatomy remains as relevant as ever (*Holliday et al., 2013*). The current emphasis on 3D modelling approaches indicates that anatomical data should be collated and communicated in 3D, as much as is practically possible. It is certainly possible to communicate anatomical information to the level required for 3D models using text or 2D diagrams; the issue is simply that a great deal of text and/or a large number of 2D images are required. Conversely, a single 3D PDF figure or web visualisation can communicate an equivalent level of information with the user being able to interact with the model to gain finer understanding of structure.

**Table 1 Jaw muscle nomenclature.** Equivalent avian jaw muscle names used in this study and selected literature. Muscle equivalents of which we are uncertain are designated by (?).

| This study | This study abbreviation | Baumel et al. (1993) | Ghetie et al. (1976) | Burton (1984) | Richards & Bock (1973) | Holliday & Witmer (2007) |
|---|---|---|---|---|---|---|
| 1. M. depressor mandibulae | mDM | M. depressor mandibulae | M. occipitomandibularis | M. depressor mandibulae | M. depressor mandibulae a, b and c | Not included |
| 2. M. adductor mandibulae externus profundus | mAMEp | M. adductor mandibulae externus rostralis temporalis | M. temporalis | M. adductor mandibulae externus rostralis temporalis | M. adductor mandibulae externus rostralis temporalis | M. adductor mandibulae externus profundus |
| 3. M. adductor mandibulae externus superficialis | mAMEs | M. adductor mandibulae externus rostralis | M. masseter superficialis (?) | M. adductor mandibulae externus rostralis lateralis | M. adductor mandibulae externus rostralis lateralis | M. adductor mandibulae externus superficialis |
| 4. M. adductor mandibulae externus rostralis temporalis post-orbital lobe | mAMErt | Part of M. adductor mandibulae externus rostralis (?) | M. masseter | M. adductor mandibulae externus rostralis temporalis post-orbital lobe | Part of M. adductor mandibulae externus rostralis temporalis (?) | Part of M. pseudotemporalis superficialis or M. adductor mandibulae externus profundus (?) |
| 5. M. adductor mandibulae posterior | mAMP | M. adductor mandibulae externus profundus caudalis | M. quadratomandibularis superficialis | M. adductor mandibulae externus caudalis | M. adductor mandibulae posterior | M. adductor mandibulae posterior |
| 6. M. pseudotemporalis superficialis | mPSTs | M. pseudotemporalis superficialis | M. mandibularis infraorbitalis (?) | M. pseudotemporalis superficialis | M. pseudotemporalis superficialis | M. pseudotemporalis superficialis |
| 7. M. pseudotemporalis profundus | mPSTp | M. pseudotemporalis profundus | M. quadratomandibularis profundus oralis | M. pseudotemporalis profundus | M. pseudotemporalis profundus | M. pseudotemporalis profundus |
| 8. M. pterygoideus dorsalis medialis anterior | mPTDma | M. pterygoideus dorsalis medialis | M. pterygomandibularis (?) | M. pterygoideus dorsalis medialis | M. pterygoideus dorsalis medialis anterior | M. pterygoideus dorsalis |
| 9. M. pterygoideus dorsalis medialis posterior | mPTDmp | M. pterygoideus dorsalis medialis | M. cranioptergygoquadratus oralis (?) | M. pterygoideus dorsalis medialis | M. pterygoideus dorsalis medialis posterior | M. pterygoideus dorsalis |
| 10. M. pterygoideus dorsalis lateralis | mPTDl | M. pterygoideus dorsalis lateralis | M. mandibulopalatinus pars lateralis | M. pterygoideus dorsalis lateralis | M. pterygoideus dorsalis lateralis | M. pterygoideus dorsalis |
| 11. M. pterygoideus ventralis medialis | mPTVm | M. pterygoideus ventralis medialis | M. mandibulopalatinus pars medialis | M. pterygoideus ventralis medialis | M. pterygoideus ventralis medialis | M. pterygoideus ventralis |
| 12. M. pterygoideus ventralis lateralis part 1 | mPTVl pt1 | M. pterygoideus ventralis lateralis | Not determined | M. pterygoideus ventralis lateralis | M. pterygoideus ventralis lateralis | M. pterygoideus ventralis |
| 13. M. pterygoideus ventralis lateralis part 2 | mPTVl pt2 | M. pterygoideus ventralis lateralis | Part of M. mandibulopalatinus pars medialis (?) | M. pterygoideus ventralis lateralis | M. pterygoideus ventralis medialis fan | M. pterygoideus ventralis |
| 14. M. pterygoideus ventralis lateralis part 3 | mPTVl pt3 | M. pterygoideus ventralis lateralis | M. entotympanicus (?) | M. pterygoideus ventralis lateralis | Not determined | M. pterygoideus ventralis |

## ACKNOWLEDGEMENTS

The authors would like to thank Phillip Clausen, Michael Mahony, Stephen Wroe, John Rodger, Walter Boles, Jaynia Sladek, Australia Museum, Eleanor Cunningham, Newcastle Calvary Mater Hospital.

### Funding

This project was funded by an Australian Research Council Discovery Project (grant number DP0986471) to CRM and through Monash University internal funding to CRM. The funders had no role in study design, data collection and analysis, decision to publish, or preparation of the manuscript.

### Grant Disclosures

The following grant information was disclosed by the authors:
Australian Research Council Discovery Project: DP0986471.

### Competing Interests

David G. Barnes is an employee of the VLSCI Life Sciences Computation Centre.

### Author Contributions

- Michelle R. Quayle conceived and designed the experiments, performed the experiments, analyzed the data, wrote the paper, prepared figures and/or tables, reviewed drafts of the paper.
- David G. Barnes analyzed the data, wrote the paper, prepared figures and/or tables, reviewed drafts of the paper, data presentation, and produced the 3D PDF model.
- Owen L. Kaluza analyzed the data, prepared figures and/or tables, data presentation, and produced the web-based model.
- Colin R. McHenry conceived and designed the experiments, analyzed the data, wrote the paper, reviewed drafts of the paper.

### Animal Ethics

The following information was supplied relating to ethical approvals (i.e., approving body and any reference numbers):

Institutional review does not apply as research was conducted upon museum specimens only.

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

## APPENDIX: 3D visualisation of skull bones and jaw musculature in *Dacelo novaeguineae*.

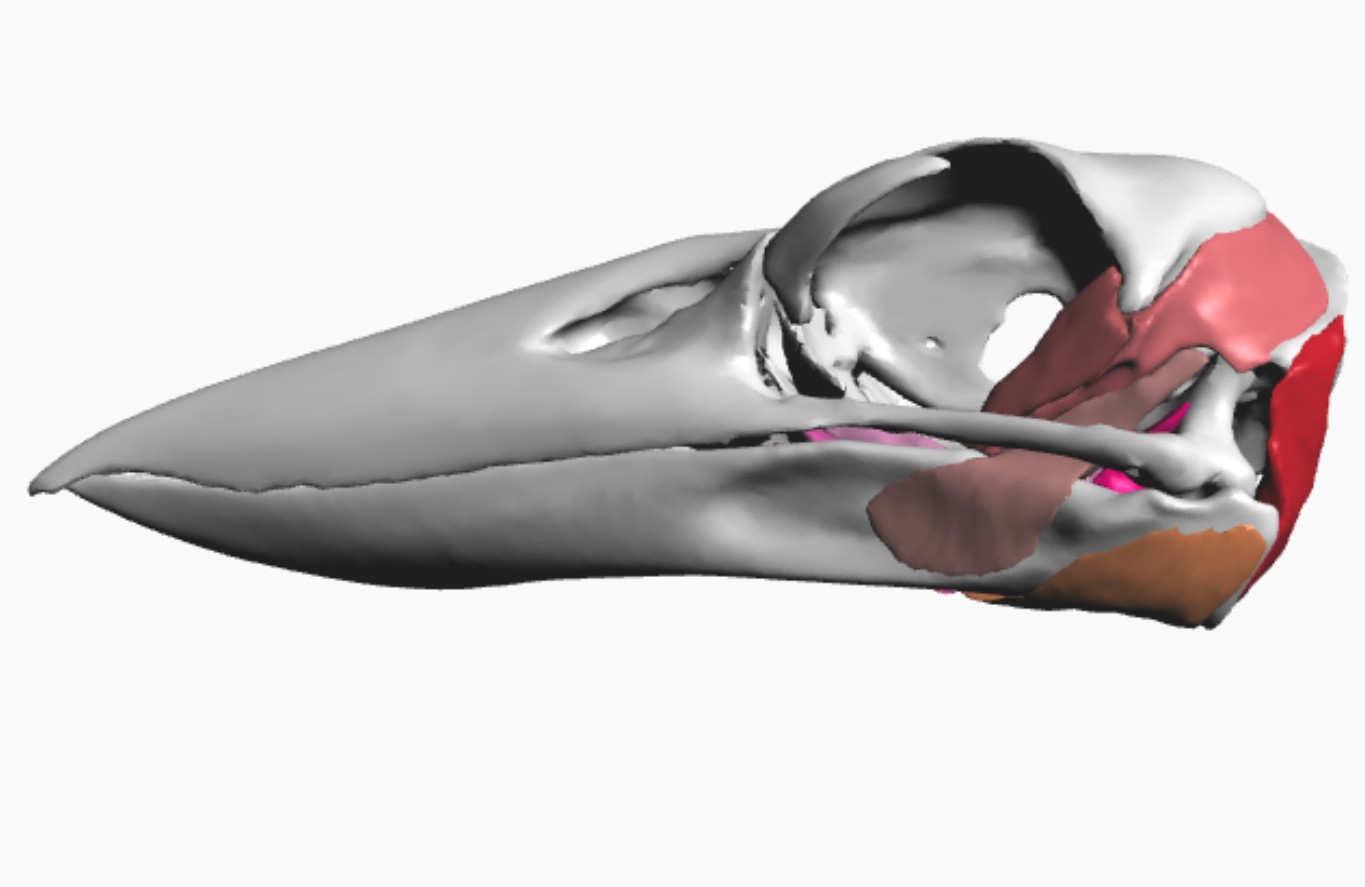

Object:

Toggle: Cranium; Mandible; Depressor muscle; Temporal muscles; Pseudotemporalis muscles; Dorsal Pterygoid muscles; Ventral Pterygoid muscles.

This 3D figure is an appendix to Quayle et al., "An interactive three dimensional approach to anatomical description - the jaw musculature of the Australian laughing kookaburra (*Dacelo novaeguineae*)." If viewing from within Adobe Reader, click on the image to activate the 3D model. To use the model, click to view the label for each structure, left click and drag to free rotate, control left click and drag to pan, use scroll wheel to zoom. Select the "Toggle Model Tree" icon to show/hide individual components of the model. Right click on the component names in the "Model Tree" to make transparent. Muscle labels: 1 mDM *M. depressor mandibulae*, 2 mAMEp *M. adductor mandibulae externus profundus*, 3 mAMEs *M. adductor mandibulae externus superficialis*, 4 mAMErt *M. adductor mandibulae externus rostralis*, 5 mAMP *M. adductor mandibulae posterior*, 6 mPSTs *M. pseudotemporalis superficialis*, 7 mPSTp *M. pseudotemporalis profundus*, 8 mPTDma *M. pterygoideus dorsalis medialis anterior*, 9 mPTDmp *M. pterygoideus dorsalis medialis posterior*, 10 mPTDl *M. pterygoideus dorsalis lateralis*, 11 mPTVm *M. pterygoideus ventralis medialis*, 12 mPTVl pt1 *M. pterygoideus ventralis lateralis* Part 1, 13 mPTVl pt2 *M. pterygoideus ventralis lateralis* Part 2 and 14 mPTVl pt3 *M. pterygoideus ventralis lateralis* Part 3.