# Peer review of "An interactive three dimensional approach to anatomical description—the jaw musculature of the Australian laughing kookaburra (Dacelo novaeguineae)"

_PeerJ, doi:10.7717/peerj.355_

## Round 0.1 · original submission · Minor Revisions

Reviewer 1 found some parts of the manuscript has been published by the authors in PLoS One-Embedding and Publishing Interactive, 3-Dimensional, Scientific Figures in Portable Document Format (PDF) Files. Please provide your original paper from Plos One in the supplement submission for the editorial check, to avoid possible overlapped publication, if any, along with a letter of justification if necessary

Reviewer 1 ·

Basic reporting

i found some parts of the manuscript is published by the authors in PLoS One-Embedding and Publishing Interactive, 3-Dimensional, Scientific Figures in Portable Document Format (PDF) Files. However, the authors added additional information, so I thought it can be accepted with minor modification.

Experimental design

I am not an expertise in this field, but what I suggest many references are material section are less only Barnes et al., . is given. So, more reference should be added. Authors should also clarify why there is requirement of 2D stracture?

Validity of the findings

No comments

Comments for the author

I hope this type of integrated approach will help to continue in the field of anatomy. I appreciated the work.

·

Basic reporting

no comments

Experimental design

no comments

Validity of the findings

no comments

Comments for the author

The authors present an anatomical model of the jaw muscles of the kingfisher Kookaboora. The paper is straightforward and generally written well.

Sundry comments:
Ln 206. The pseudotemporalis is attaching to the laterosphenoid here. “Orbitosphenoid” is somewhat antiquated and there remains a lot of confusion regarding its homologies between reptiles and birds. Also, when there is a clear orbitosphenoid ossification, it is typically on the most medial part of the orbital surface near the optic canal... rather than laterally where this muscle attached. This is all of course challenging in birds which typically lack clear sutures in the braincase. But the laterosphenoid and pseudotemporalis superficialis muscles are basically married to one another throughout archosaur evolution, including non-avian dinosaurs, so I would stick with "laterosphenoid".

The orbitosphenoid process is ok as a structure name (though see above). But this is really cool: basically the zygomatic and orbitosphenoid processes here are bony spurs for the tendon attachments for the respective jaw muscles. Zusi and Livezey 2000 worked all this out (citation below) in galloanseriforms, but no doubt it may apply to kingfishers. I’ve seen similar structures in grebes. So you may want to be careful actually assuming these are homologous with the lower temporal bar (i.e., zygomatic process; which it’s not), or then extensions of the orbitosphenoid bone. My hard copy of this paper has been destroyed, and I’ve failed to get a digital copy of it in the past several days to check the terminology they suggest.

Zusi, Richard L. and Livezey, B. C. 2000. Homology and phylogenetic implications of some enigmatic cranial features in Galliform and Anseriform birds. Annals of Carnegie Museum, 69(3): 157-193.

Ln 219 need space
Ln 222 I don’t think maxillopalatine is an appropriate term (there certainly isn’t a bone of this name).
Ln 303. Much of this paragraph isn’t needed. “m.” or “M.” is the abbreviation for “musculus”, so most authors use the latter once and then the abbreviation. Different authors/journals/editors prefer m. vs M. as well as whether or not the names are italicized in the text. There is no “right” way although consistency is of course nice albeit impossible. So...much of this is a non-issue and the paragraph could be removed.

Fig 1 caption. Species name needs to be italicized.

---

## Round 0.2 · Minor Revisions

Thank you for your revision. Reviewers #4 and #5 in this email were the individuals who looked at the original submission. During the period when your revision is evaluated, I took the advice of some additional reviewers and in particular reviewer 1 had some very pertinent observations. In order to improve the overall quality of the manuscript, please carefully address the concerns raised.

Reviewer 1 ·

Basic reporting

1. The style of the reference should be formatted according to the PeerJ Instructions for Authors. For example, in the in-text citations, the format should be “First author et al., year” for four or more authors, the authors of this manuscript didn’t use a comma “,” after “et al.” for all of the in-text citations of this type. In addition, the volume numbers should be bolded.

2. Some grammar and spelling mistakes need to be corrected, as shown below:
a. Abstract, line 7: in “We identified 14 major jaw muscles; 6 in the temporal group……”, the semicolon “;” should be changed to colon “:”
b. Discussion, lines 379 and 380: in “with the user able to interact with the model to gain finer understanding of structure”, “being” should be added between “user” and “able”
c. The authors used “visualization” and “visualisation” interchangeably throughout the manuscript, although they are both correct as one is American usage while the other one is British usage, it is better to use only one of them within one article to keep consistency.
d. References, lines 449 and 450: “Quayle MR. 2011. A Morphological Study of the Kingfisher Skull Environmental Science and Management Honours. University of Newcastle.” This reference seems cited incorrectly.

Experimental design

Why Dacelo novaeguineae was picked for study using the interactive 3D visualization approach? The authors mentioned in the Introduction section that “The following description of the jaw musculature of the Australian laughing kookaburra (Dacelo novaeguineae) was stimulated by the need to incorporate detailed anatomical data into high resolution 3D finite element models as part of an analysis of skull biomechanics in 8 species of kingfisher (Alcedines) (Quayle 2011; Quayle et al. in prep.)”, however, a search for these two references “Quayle MR. 2011. A Morphological Study of the Kingfisher Skull Environmental Science and Management Honours. University of Newcastle” and “Quayle et al. in prep.” could not lead to any useful information about the purpose of this study. It is recommended that the authors should either check the accuracy of the citation of the first “Quayle 2011” reference or clearly state concisely why Dacelo novaeguineae was picked for this study.

Validity of the findings

No Comments.

Comments for the author

This manuscript described an interactive three dimensional approach to the anatomical description of the cranial musculoskeletal anatomy of the Australian Laughing Kookaburra, Dacelo novaeguineae, with a focus on the geometry and attachments of the jaw muscles in this species. Fourteen major jaw muscles were identified in this manuscript, of which the 3D visualization model was presented in a figure. The interactive 3D visualization represents a relatively new approach to the anatomical description that would contribute to the accurate documentation and communication of detailed anatomical structures as well as the study on the relationship between form and function in biology. In conclusion, this manuscript is recommended to be published in PeerJ after major revisions.

Besides the aspects that have been already mentioned in the Basic Reporting and Experimental Design sections above, two other issues that need to be addressed are shown below:

1. Throughout this manuscript the interactive 3D visualization approach was highlighted, along with a representative 3D visualization of the skull bones and jaw musculature of Dacelo novaeguineae presented in Figure 5 in the Appendix. However, for such an important 3D visualization model, the authors used the same picture that has been published in PLoS One in September 2013 in a paper titled “Embedding and Publishing Interactive, 3-Dimensional, Scientific Figures in Portable Document Format (PDF) Files” by David G. Barnes, et. al. Even if Plos One does not retain copyright of the published material, this is called “self-plagiarism”. Without any annotation, the exactly the same picture that has been published previously should not be included in a manuscript intended to be published as an original research article. It is recommended that the authors should use another picture of the skull bones and jaw musculature of Dacelo novaeguinea processed by the interactive 3D visualization approach.

2. The authors mentioned very simply that “This method of interactive visualisation is becoming increasingly popular, with publications in biomedical science (Kumar et al. 2009; Selvam et al. 2009), astrophysics (Barnes & Fluke 2008), invertebrate (Baeumler et al. 2008; Neusser et al. 2009; Ziegler et al. 2008) and vertebrate anatomy (de Bakker et al. 2012; Holliday et al. 2013; Ruthensteiner & Heß 2008) and paleontology (Knoll et al. 2012; Lautenschlager 2013).” in the Introduction section, and discussed “the role of 3D images in communicating anatomical information” in the Discussion section. However, it would be better if some more information on the significance and application of this interactive 3D visualization approach (e.g., how can this method be generalized to other subjects and what types of important information can be provided by this method) is added at the end of the Discussion section.

Reviewer 2 ·

Basic reporting

The author clearly identified 14 major jaw muscles of Australia Laughing Kookaburra Dacelo novaeguineae, with the summary of nomenclature and 3D model. This paper is well written, and the 3D model is novel and easy to understand.

Experimental design

No comments

Validity of the findings

No comments

Reviewer 3 ·

Basic reporting

No Comments

Experimental design

Is there any other ways to digitally analyzed the skull of adult laughing kookaburra except clinical computed tomography (CT) scanner, with a higher resolution?

Validity of the findings

1) Is this specimen representative? What's the age of the deceased laughing kookaburra, and whether this age of the kookaburra is able to represent an adult common kookaburra? It is better for the authors to describe more detailed on the specimen seletion.

2)To make it statistically significant, is one sample enough?

3)The authors may want to discuss more detailed on the data compared with other previously reported kookaburra skull and muscle structures, especially muscle attachment points, to make their data and results more convincing and representative?

Comments for the author

May want to discuss more on comparing with previously reported skull and muscle structures.

Reviewer 4 ·

Basic reporting

No comments

Experimental design

No comments

Validity of the findings

No comments

Comments for the author

Happy to declare that the manuscript is significantly improved and the results presented could be useful for the reader of this journal.

Reviewer 5 ·

Basic reporting

The revisions have fixed the issue I found.

Experimental design

no comment

Validity of the findings

no comment

Comments for the author

no comment

---

## Round 0.3 · Minor Revisions

I would suggest that the author should carefully address the reviewers' concerns, especially for those that has not been well addressed in the previous version.

Please provide more details on how to obtain the thesis regarding "whilst the other is a honours thesis which is available from the University of Newcastle or from the author upon request").

In addition, since the 3D model method has already been published in another paper, to make this paper better, is it possible to clarify the function meaning of 14 jaw muscles as well as to make a comparison in regards to muscle function between other birds’ jaw muscles? accordingly, if possible, could you perform a thorough literature search or use the Pubmed resources.

Reviewer 1 ·

Basic reporting

The authors have corrected the style of the references and some grammar and spelling mistakes based on my previous comments.

Experimental design

Regarding my previous question “Why Dacelo novaeguineae was picked for study using the interactive 3D visualization approach?”, the authors replied that they feel the sentence in the manuscript “The following description of the jaw musculature of the Australian laughing kookaburra (Dacelo novaeguineae) was stimulated by the need to incorporate detailed anatomical data into high resolution 3D finite element models as part of an analysis of skull biomechanics in 8 species of kingfisher (Alcedines) (Quayle 2011; Quayle et al. in prep.)” already describes the reason. However, as mentioned by the authors, “the reviewer could not find the mentioned references as one is “in prep” meaning in preparation and so not yet published, whilst the other is a honours thesis which is available from the University of Newcastle or from the author upon request.” Due to the difficult access to the relevant references, I still recommend that the authors should explain clearly why Dacelo novaeguineae was picked for the present study for the following two reasons:
1. Not every reader has background information about Dacelo novaeguineae, then they need information to understand why there is a need for detailed anatomical data of Dacelo novaeguineae cranial musculoskelet
2. As mentioned by the authors, the PLos ONE paper focused on the demonstration of “the open source code that we have used to generate interactive 3D figures that can be embedded into PDF documents, and show its applicability to different disciplines (including, in that paper, anatomy, neuroscience, biochemistry, and astronomy)”. Since this interactive three dimensional approach itself was the focus of the PLos ONE paper while the present paper focused on the anatomy of Dacelo novaeguineae, it is necessary to explain why Dacelo novaeguineae was picked for the 3D anatomical study, to clarify the significance and importance of the present study, and also to make this study distinguished from the PLos ONE paper.

Validity of the findings

No comments.

Reviewer 2 ·

Basic reporting

No Comments

Experimental design

It will be better to elucidate that the sample is able to represent as a typical model of adult Australian Laughing Kookaburra. The authors may want to provide more information of this Kookaburra skull. For example: gender, age and probably detailed appearance description of it.

Validity of the findings

The authors may also want to emphasize what is the difference between their nomenclature and previous published nomenclature. And why their study is important in this field.

Comments for the author

No Comments

---

## Round 0.4 · accepted · Accept

I am pleased to inform that your manuscript has been accepted.

Reviewer 1 ·

Basic reporting

The authors have already made corrections according to my previous two rounds of reviews.

Experimental design

The authors have already made changes based on my previous reviews.

Validity of the findings

No Comments.

Reviewer 2 ·

Basic reporting

No comments

Experimental design

No comments

Validity of the findings

No comments

Comments for the author

No comments